# Electrochemical Impedance Spectroscopy (EIS) Explanation of Single Crystal Cu(100)/Cu(111) in Different Corrosion Stages

**DOI:** 10.3390/ma16041740

**Published:** 2023-02-20

**Authors:** Qihao Lin, Guoqing Chen, Shiwen Zou, Wenlong Zhou, Xuesong Fu, Shuyan Shi

**Affiliations:** 1Key Laboratory of Solidification Control and Digital Preparation Technology (Liaoning Province), School of Materials Science and Engineering, Dalian University of Technology, Dalian 116085, China; 2Dalian Technology (Yingkou) Advanced Material Engineering Center Company Limited, Yingkou 115004, China; 3Aerospace Research Institute of Materials and Processing Technology, Beijing 100000, China; 4School of Materials Science and Engineering, Dalian University of Technology, Dalian 116085, China

**Keywords:** EIS, anisotropy, single crystal, equivalent electrical circuit

## Abstract

Copper and its alloys are used widely in marine environments, and anisotropic corrosion influences the corrosion kinetics of copper. Corrosion of copper in an electrolyte containing ${\mathrm{Cl}}^{-}$ is described as a dissolution–deposition process, which is a prolonged process. Therefore, it is laborious to clarify the corrosion anisotropy in different stages. In this paper, electrochemical impedance spectroscopy (EIS) following elapsed open circuit potential (OCP) test with 0 h (0H), 24 h (24H) and 10 days (10D) was adopted. To exclude interruptions such as grain boundary and neighbor effect, single crystal (SC) Cu(100) and Cu(111) were employed. After 10D OCP, cross-sectional slices were cut and picked up by a focused ion beam (FIB). The results showed that the deposited oxide was Cu_2_O and Cu(100)/Cu(111) experienced different corrosion behaviors. In general, Cu(100) showed more excellent corrosion resistance. Combined with equivalent electrical circuit (EEC) diagrams, the corrosion mechanism of Cu(100)/Cu(111) in different stages was proposed. In the initial stage, a smaller capacitive loop of Cu(111) suggested preferential adsorption of ${\mathrm{Cl}}^{-}$ on air-formed oxide film on Cu(111). Deposited oxide and exposed bare metals also played an important role in corrosion resistance. Rectangle indentations and pyramidal structures formed on Cu(100)/Cu(111), respectively. Finally, a perfect interface on Cu(100) explained the tremendous capacitive loop and higher impedance (14,274 Ω·cm^2^). Moreover, defects in the oxides on Cu(111) provided channels for the penetration of electrolyte, leading to a lower impedance (9423 Ω·cm^2^) after 10D corrosion.

## 1. Introduction

Corrosion of copper and its alloys containing ${\mathrm{Cl}}^{-}$ environment has attracted great attention due to the excellent corrosion resistance to seawater and wide usage in marine environments, such as in the manufacturing of pipes, pumps, valves and ship propellers [1]. The corrosion behavior of copper in the ${\mathrm{Cl}}^{-}$ environment is described as a dissolution and deposition process [2,3,4]. Zhao [5] used a dissolution–ionization–diffusion–deposition (DIDD) process to define the complex corrosion of 20 steel and 3Cr steel. In this mode, various types of metal atoms showed different dissolution rates in the M/S interface when a metal atom dissolved and formed a cation. Subsequently, different concentrations of cation led to various levels of hydrolysis. Consequently, the supersaturation degree and critical surface energy of nucleation for corrosion product would be changed. During the prolonged corrosion process, the anisotropic corrosion behavior of copper was found in the dissolution and deposition stages since atomic density changes in different orientations [6]. Moreover, extensive literature indicated that corrosion behavior strongly depends on the local surface reactivity, and the surface reactivity itself can be strongly influenced by the crystallographic orientation [7,8,9]. X.Y. reported that the orientation-dependent corrosion rate of 90Cu-10Ni in acidic NaCl solution followed the order of (001) < (101) < (111) [10]. Ma [11] proposed a surface step dissolution model to explain the faster dissolution of the closest packed (111) planes, where coordination number played an important role in understanding the abnormal corrosion rate. Evenly in pure water, fine copper wires showed anisotropic dissolution where the dissolution rate increases in the order of (111) < (001) < (110) [12]. During the deposition, Cu_2_O is one of the possible products after the corrosion of copper in NaCl solutions, according to the Pourbaix diagram. Cu_2_O shows the capacity of a P-type semiconductor [13,14], which means some protection from penetration. Furthermore, the protective ability of oxides is associated with the compactness of the crystal. It was reported that granular oxides on Cu(100) contributed to the faster dissolution of oxides [15,16,17]. Therefore, less compact oxide is unhelpful, although passivation can be predicted by the Pourbaix diagram.

Through the above analysis, anisotropic corrosion varied in different stages. Moreover, electrochemical impedance spectroscopy (EIS) is a powerful tool in the investigation of corrosion behavior that not only provides a non-destructive assessment of the corrosion rate but also enables the determination of the corrosion mechanism [18,19]. EIS can separate the diffusion processes from other physico-chemical processes in a wide frequency spectrum, which has served as a powerful and routine tool in estimating the diffusion coefficient [20,21]. Previously, samples were cleaned and migrated from a corrosive environment to an electrochemical workstation after corrosion [22,23]. In this way, the corrosion frontier, namely the solution/metal (S/M) interface, was changed, and the new balance was reconstructed at the interface instead of the S/M interface in the original corrosive conditions. In this paper, immediate EIS following elapsed open circuit potential (OCP) test was adopted.

Microstructures, such as grain boundaries [24,25], twins [26,27] and grain size [28,29], can also influence corrosion behavior profoundly. At present, it has been reported that neighbor grain also laid an effect on corrosion [30]. Single crystal (SC) [31,32,33] has some advantages in excluding various influences of microstructures. For example, P.B. [34] successfully confirmed the anisotropic dissolution of Fe_3_Al SCs in sulphuric acid. Therefore, (100) planes of Fe_3_Al with the highest atomic density showed a modest corrosion rate. L.N. [33] also reported that a well-compact passive film with field-controlled formed on (013)/(014) planes and a porous film with diffusion control formed on (146) planes. Therefore, SC Cu(100)/(111) were adopted in this paper.

As a result, the purpose of this study is to elucidate the corrosion anisotropy of copper in different stages by EIS of SC Cu(100)/Cu(111). SCs were immersed in 3.5 wt.% NaCl solutions under OCP monitoring for 0 h (0H), 24 h (24H) and 10 days (10D), followed by EIS immediately. SCs exhibited various behaviors in different corrosion stages, and differences were also found between Cu(100) and Cu(111) for the same corrosion time. Bare metals and oxides on SCs were both taken into consideration to explain the variations in corrosion behaviors. Distinct interface for Cu(100)/Cu(111) was an important factor contributing to the protective ability of deposited oxides. Combined with an equivalent electrical circuit (EEC), the corrosion mechanism of SCs in different stages was discussed finally.

## 2. Materials and Experimental

SC Cu(100)/Cu(111) plates (≥99.99% purity, mass%) were supplied by Yuanjing Technology Corporation at Hefei of China, of which dimensions were 10 × 10 × 1 mm^3,^ and misorientation was less than 2°. The crystallographic orientation of SCs was checked out by electron backscattered diffraction (EBSD, SUPARR 55, Zeiss, Germany) and X-ray diffraction (XRD, Empyrean, PANalytical B.V., Almelo, The Netherlands) with Cu Ka radiation before immersion. The inverse pole picture (IPF) in Figure 1a,b, as well as the presence of (200) and (111) peak in Figure 1c, elucidated the preferential orientation of SCs that normal directions were parallel to 〈100〉 and 〈111〉 direction, respectively.

SCs were polished with 0.4 μm silica sol and cleaned with acetone and alcohol to eliminate contamination before prolonged OCP testing. The electrolyte was 3.5 wt.% NaCl solutions mixed with reagent grade NaCl and deionized (DI) water with a resistance of more than 18.2 MΩ·cm. The OCP testing was conducted for 0H, 24H and 10D, respectively, followed by EIS immediately. A standard three-electrode system with a platinum grid as the counter electrode, a KCl-saturated calomel electrode (SCE) as the reference electrode and Cu SCs as the working electrodes was applied on CS310 electrochemical workstation by CorrTest Corporation at Wuhan of China. Electrochemical impedance spectroscopy (EIS) tests were conducted by setting a set alternating current (AC) signal of 10 mV to a frequency range of 10^5^ Hz to 10^−2^ Hz. The results of EIS were analyzed and simulated by ZView 2 embedded in CS studio 5 software. A potentiodynamic scan was conducted from −0.8 V to 0.8 V (vs. SCE) at a rate of 1 mV/s, and the partial range was shown in this paper. The morphologies of corrosion products (CPs) after immersion in electrolytes were imaged on scanning electron microscopy (SEM, JSM-7610F Plus, JEOL, Tokyo, Japan) and assembled with a backscatter (BSE) detector. Phase constitution was detected by energy dispersive spectroscopy (EDS, Oxford Instruments, Abingdon, UK) detector, XRD, as well as electron probe microanalysis (EPMA, JXA-8530F PLUS, JEOL, Japan). Three-dimensional (3D) topography of corroded samples was obtained by laser-scanning confocal microscopy (LSCM, OLS4000, Olympus, Tokyo, Japan). To acquire the cross-sectional information of corroded samples, slices at different zones were milled and picked up by a focused ion beam (FIB, Helios G4 UX, Thermo Fisher, Waltham, MA, USA), of which the thickness was approximately 4 μm enough for SEM and EDS characterization.

## 3. Results and Discussion

### 3.1. Morphologies and Phase Constitution after 24H and 10D OCP

The morphologies of corroded samples were displayed in Figure 2 after different durations of OCP tests. Many rectangular indentations in different sizes were observed in Figure 2a after 24H OCP. The side walls of the indentations were vertical to the sample surface, which was like the embedded diagram at the lower-right corner. Other profiles can be viewed as a combination of adjacent smaller rectangles. The enlarged picture in Figure 2a showed that the bottom was not a flat plane but with extensive steps, and the corners of steps approximately equaled 90°. Morphology changed drastically after 10D corrosion of Cu(100) SC, as shown in Figure 2b. Two types of features accounted for the corroded surface, namely rough flocculent-like region and slightly flat planes with extensive pits, which means corrosion products are completely covered on the substrate. For Cu(111) SC after 24H OCP shown in Figure 2c, many inverted pyramids were detected, and the substrate was exposed at the bottoms of pyramids. Many shaded triangles were also exhibited under BSE observation, which should be treated as component segregation, and further EPMA results would confirm this. After 10D OCP, the corroded surface of Cu(111) showed a slushy appearance with some small flakes on it, and the pyramidal structures totally disappeared. From the vague tripe (surrounded by red dotted lines) and enlarged morphology in Figure 2d, the flakes on Cu(111) might be the remained surface, which was like the flat areas in Figure 2b. It was deduced that more extensive dissolution occurred on Cu(111), and the pyramidal structures were covered totally by corrosion products. X.Y. [10] deduced that (001), (111), (101) and (112) planes were all possible faceted planes depending on the deviated angle against low-indexed planes. Herein, the vertical side walls in Cu(100) SC and inverted pyramids, which can be viewed as one corner cut by (111) planes showed in Figure 2c, suggested that the faceted planes are (100) planes.

The constitution of different positions after 24H OCP was detected by EPMA, and results were generated in Table 1. Position 1 was at the bottom of the corroded pit, and almost complete Cu accounted for the constitution with a slight O element. It was suggested that these rectangular pits remained by a dissolution process, and the bare Cu was exposed at the bottoms of the pits. Furthermore, the atomic ratio of Cu/O exceeded that of Cu_2_O, which is a reasonable corrosion product of Cu in 3.5% wt.% NaCl solutions. It might be the result that the thickness of the oxide was too small, and the electron beam detected the substrate across the surficial oxide; it was different for Cu(111) SC. The composition of the side wall at position 3 was almost Cu, while the constitution at position 4 approached Cu_2_O, which means that the depth of oxides in the triangles was larger than that of oxides in Cu(100) SC. Composition at position 5 was similar to position 2 but with a slightly higher atomic ratio of Cu/O, which can be explained by the fact that thicker oxides formed on Cu(111) SC after the same corrosion time. The component of corrosion products was further characterized by XRD after 10D OCP. Compared with the single peak for as-received SC, the square frames A and B in Figure 1c were enlarged views of Cu(100)-10D from 40° to 45° and Cu(111)-10D from 35° to 40°, where slight Cu_2_O(200) and Cu_2_O(111) peaks were detected. The subsequent peaks confirmed the reliability of the above analysis that the constitution of oxide was Cu_2_O. It was noted that ${\mathrm{Cl}}^{-}$ level remained stable, approaching 0, which should be viewed as background noise and can be neglected.

To compare the thickness of oxides on different surfaces, cross-sectional slices were cut and picked up by FIB, which included the substrate, oxides and deposited Pt layer. Figure 3a and Figure 4a showed the sectional appearances from flat and flocculent-like regions on Cu(100) SC, respectively. From the analysis above, the surface was smooth with several indentations, and the indentations did not stretch to the substrate in Figure 3a. The interface between oxides and Cu was composed of several segments. Interestingly, some of the segments showed a degree against the horizon instead of being vertical, as the red dotted lines show. The interface was almost perfect with rare leaks, and the thickness of the oxides layer ranged from 0.7 μm to 0.9 μm. However, Figure 4a shows an uneven surface and more defects at the interface. The enlarged picture in the middle upper part suggested the leak was blocked and did not exist at a long distance. Moreover, the connecting parts at the interface were compact, and the thickness of oxides varied at a larger range from 0.5 μm to 1.5 μm. In Figure 3b, a less compact interface was observed for Cu(111) SC situation, where long-distance leaks existed at the interface and crevice stretched along the whole interface. It departed the substrate from oxides obviously. Furthermore, leaks also existed inside the oxides, as the enlarged pattern shows, which means terrible cohesion between oxides and substrate. It may contribute to the easy penetration of electrolytes into the substrate.

The constitution of oxide was also characterized by an EDS map, line and point scan, as shown in Figure 4. Element map showed good consistency with SEM profiles, where O highlighted the intermediate layer and Cu changed inapparently. ${\mathrm{Cl}}^{-}$ served as background noise, which was further verified during a line scan with an extremely low signal. The signal of O remained steady at a compact range of oxides, and the climbing curve of Cu may be caused by the interruption of deposited Pt. Finally, the mass ratio of Cu/O was 9/1, which was slightly higher than Cu_2_O. It can be explained as the deviation of EDS or incomplete oxidation during corrosion. According to the Pourbaix diagram [2,4,35,36] and extensive research [3,37], ${\mathrm{CuCl}}_{2}^{-}$ is believed to be the main cuprous complex in seawater and NaCl electrolytes, with approximately 0.55 mol/L concentrations of ${\mathrm{Cl}}^{-}$. ${\mathrm{CuCl}}_{2}^{-}$ was deposited on the substrate according to Formula (1) in the condition where supersaturation was achieved. By combining the constitution analysis above, the oxides can be defined as Cu_2_O, and similar EDS results were not displayed repeatedly in this paper.
(1)$$2{\mathrm{CuCl}}_{2}^{-}+2{\mathrm{OH}}^{-}↔{\;\mathrm{Cu}}_{2}O+{\;H}_{2}O+4{\mathrm{Cl}}^{-}$$

Three-dimensional morphologies (with quintuple height factor) after 10D corrosion were also detected, as shown in Figure 5. In the condition where the same scale range was selected, oxides on Cu(111) SC showed higher protrusions than the plentiful red region shown in the 3D map. The calculated surface roughness $S_{a}$ were 0.162 μm and 0.189 μm for Cu(100) and Cu(111), respectively. The sectional profiles were further plotted, and the surficial outline of Cu(111) varied at a larger range than that of Cu(111), as the red curves show. The results of sectional linear rugosity $R_{a}$ for Cu(100) and Cu(111) were 0.126 μm and 0.154 μm, respectively. All the information above inferred that a more rugged surface was constructed on Cu(111) by deposited Cu_2_O.

### 3.2. EIS Explanation in Different Corrosion Stages

Different durations of OCP were conducted, and the sustained 10D OCP of Cu(100)/Cu(111) SCs were recorded, as shown in Figure 6. Both curves climbed sharply to peaks and then decreased slowly. Finally, OCP fluctuated around −0.24 V for Cu(100) and −0.25 V for Cu(111), separately. The final steady OCP of an electrode boils down to the dynamic equilibrium dissolution between the formations of oxide film on the working electrode surface in a specific electrolyte [38]. When the OCP is more positively stabilized, it symbolizes that the sample has a lower tendency to corrode, whereas the less positively stabilized OCP value suggests the opposite [39]. More negative OCP for Cu(111) indicated unstable oxides on the electrode. According to SEM morphologies and constitution above, Cu_2_O formed immediately on SCs, which led to the obvious increase in potential in the initial stage. Both SCs reached to peak at around 24H, but the OCP of Cu(111) decreased at once when the maximum arrived. This might be the less stability of oxides on Cu(111), as the coarser surface in Figure 5b shows. The higher potential of Cu(100) at last can also prove well protection of oxides on it.

The Nyquist diagrams of Cu(100)/Cu(111) SCs after different OCP periods were plotted in Figure 7a, where different geometric polygons represented the experimental states and the blue lines symbolized the fitting results. In the initial corrosion stage, a depressed capacitive loop from high frequency to medium frequency and an incomplete capacitive loop in low frequency defined the Nyquist diagram of Cu(100) and Cu(111) SCs. Moreover, the loop’s radius of Cu(100) was larger than that of Cu(111), which indicated a more protective air-formed film on Cu(100). As corrosion time prolonged, both Cu(100)/Cu(111) SCs obtained a straight line in low frequency [19,40]. As it is known, the straight line was usually defined as the Warburg impedance, which is attributed to the mass transport in the corrosion reaction [41,42]. A perfect planar electrode induced a degree of 45° against the axis; moreover, a coarse electrode would reduce the inclination, as shown in Figure 7a. It was noted that the radius of the capacitive loop in high frequency showed contradicting variation, namely an increase for Cu(111) SC and a decrease for Cu(100) SC. This alteration must be associated with the dissolution–deposition process. After 10D corrosion, a large arc accounted for the Nyquist diagram in the whole frequency range for Cu(100) SC. While for Cu(111) SC, the radius of the capacitive loop in high frequency further increased, and the straight line disappeared, which was substituted by an incomplete arc in low frequency. It was deduced that the diffusion was depressed by deposited oxides on Cu(111) SC.

In general, Cu(100) SC showed a larger radius of the capacitive loop than Cu(111) SC in different corrosion stages, which was opposite to conventional surface energy theory [11]. It was said that (111) planes in face-centered cubic (FCC) metals with the highest atomic density showed excellent corrosion resistance. However, E.M. [8] and S.O. [12] thought that air-formed oxide film was responsible for corrosion resistance instead of bare metals. In electrolyte containing ${\mathrm{Cl}}^{-}$, the dissolution process is strengthened due to the prototypical activator of ${\mathrm{Cl}}^{-}$ [43]. Thus, the reaction of ${\mathrm{Cl}}^{-}$ and Cu_2_O contributed to the abnormal corrosion behavior of Cu(100)/Cu(111) in this paper. Furthermore, G.D [44] suggested that the larger average adsorption energy of ${\mathrm{Cl}}^{-}$ on Cu_2_O {111}-type surfaces accelerated the dissolution of Cu_2_O. Therefore, a more drastic dissolution of air-formed Cu_2_O on Cu(111) explained the smaller radius of the capacitive loop. The opposite variation after 24H OCP might be the combination of exposed substrate and deposited oxides. From Figure 2, the dissolution process led to plenty of pits on Cu(100) and oxides in other areas. Additionally, Cu(111) showed a different style in that much dissolution occurred under the surface instead of exposing the substrate totally. It was reasonable to infer that the partially dissolved oxides protected the underlying substrate. Oxides formed in other areas also contributed to the increased capacitive loop on Cu(111). By contrast, uniformly distributed pits on Cu(100) without any protection showed a decrease in the capacitive loop. Finally, capacitive loops of both SCs increased, and the loop’s radius of Cu(100) was larger than that of Cu(111) at a considerable margin after 10D corrosion. This can be predicted that oxides with different appearances formed on both SCs from surficial and sectional morphology. The much larger capacitive loop without a tail in low frequency symbolized a compact oxides layer combined tightly with the substrate, shown by the almost perfect interface in Figure 3a. The separation of oxide and substrate at the interface was deleterious to corrosion resistance. Electrolytes can directly damage substrate as soon as electrolyte passes through the defects in oxides. A more rugged surface might also reduce the stability of deposited oxides due to the larger superficial area exposed in the electrolyte. The corrosion resistance for samples can be evaluated with absolute impedance (${\left|Z\right|}_{f=0.01\;\mathrm{Hz}}$) at a low frequency [38,39,45] (by bode magnitude plots (Figure 7b)), and the exact values were extracted as a histogram in Figure 7d. From the histogram, the impendence of Cu(100) were unambiguously larger than that of Cu(111) in different stages consistent with the variation in the capacitive loop in Nyquist, especially Cu(100) after 10D OCP with an impedance of 13,115 Ω·cm^2^.

A potentiodynamic scan was supplied finally to validate the electrochemical behavior during EIS disturbance, as shown in Figure 8. Corresponding surficial morphologies of corroded samples were also displayed. From the polarization curves, Cu(111) SC obtained higher OCP potential ($E_{p}$) and smaller corrosion current density ($i_{\mathrm{corr}}$) compared with Cu(111) SC. One more obvious distinction was that a narrow passive range was detected during the positive zone of Cu(100) SC. It was noted that the drastic dissolution during the polarization scan uncovered the faceted planes sharply. The hierarchical rectangular steps led to the inverted quadrangular pyramid on Cu(100) SC. A macroscopic pyramid can be visualized on Cu(111) SC, which confirms the reliability of the faceted (100) planes. The air-formed film was reduced, and bare metals were exposed during the polarization of the negative zone. As a result, a higher OCP of Cu(111) SC was observed as conventional surface energy theory predicted. Cu(111) planes with the highest atomic density showed smaller $i_{\mathrm{corr}}$ in the active dissolution zone. However, Cu(100) SC showed slight passive capability, which may be caused by more compact Cu_2_O on it. This was consistent with larger impendence in EIS and perfect interface in sectional morphology.

### 3.3. Illustration of Corrosion Mechanism by EEC

During EIS disturbance, the electric current is uniformly distributed over the entire surficial area in high frequencies, presenting the response of the whole surficial area. However, the current concentrated at the pores in low frequencies, presenting active corrosion at defects [20]. Furthermore, the broadening of the phase angle curve in medium frequencies was probably the result of time constants overlapping [18,46], which resulted in only a one-time constant. Accordingly, the empirical data obtained from EIS measurements were fitted suitably with the EECs in Figure 9. The extracted data from EIS are listed in Table 2. For easy recognition, the abbreviation of ECs of different EIS was also summarized in Table 3. The EEC of Cu(100) SC was plotted, as shown in Figure 9a, where R_s_ represented the resistance of the solution and the high-frequency circuit C_o_ symbolized the capacitance of the air-formed oxide film. The parameter groups R_ct_/C_dl_ represented double-layer capacitance and charge transfer resistance. Compared with normal two capacitances [47], the resistance of air-formed oxide film (R_o_) was absent. It is inferred that R_o_ was too small to be detected due to the ultrathin air-formed oxide film, which was usually reported as tens of nanometers. Unlike the compact Al_2_O_3_ film [21], the protection of oxide film on copper was less efficient in retarding the penetration of electrolytes. As a result, the R_ct_/C_dl_ was detected at the interface between oxide and substrate as 4548 Ω·cm^2^ and 2.89 × 10^−4^ μF·cm^−2^, respectively. Cu(111) SC used the same scenario to describe the interface response but with a decreased R_ct_ and a larger C_dl_. R_ct_ is an important parameter that is directly dependent on active protection [48]. The value of R_ct_ was usually inversely proportional to the corrosion rate of the test sample [28,49]. Smaller R_ct_ suggested that a more drastic charge transfer occurred on Cu(111) SC. This process can be illustrated in Figure 10a,d, where ${\mathrm{Cl}}^{-}$ preferentially adsorbed on the air-formed film on Cu_2_O {111}-type planes. Yan [50] reported that the initial orientation-dependent corrosion behavior was linked to the surface equivalent atomic packing density. The different protective abilities of the passive films resulted in different degrees of damage. K.H [15] suggested that after initial corrosion, the atomic composition of the (111) pyramidal plane was thought to be a combination of the (100) basal plane and the (011) diagonal plane. As a result, the equivalent corrosion tendency of the (111) oriented plane was higher than that of the (001) oriented plane. Herein, the oxides formed easily on SCs as the abrupt increase in OCP in the initial stage (Figure 6). Therefore, the corrosion resistance in this paper should be closely related to the air-formed film or deposited oxides throughout the whole corrosion process. This specific adsorption on Cu(111) caused the premature dissolution of air-formed film, which was well consistent with a smaller capacitive loop in Nyquist.

After 24H OCP, both SCs exhibited diffusion behavior as predicted. The bare substrate was exposed, as Figure 2a,c shows. Figure 10b,e explained the increased impendence of Cu(111) after 24H corrosion. Some dissolution occurred through the leaks of deposited oxides rather than complete exposure of the substrate. The corresponding EC was plotted, as shown in Figure 9b. The extracted R_ct_ for Cu(100) SC was approximately twice that of Cu(111) SC. However, if only bare substrate was considered, the latter would acquire larger resistance due to the largest atomic density. It was inferred that deposited oxides started to play a dominant role after 24H corrosion, and more compact oxides on Cu(100) SC contributed to fine resistance. Furthermore, the larger diffusion impendence Z_w_ (7071 Ω·cm^2^·s^−1/2^) of Cu(111) SC may be caused by the more tortuous channels in the thicker oxides on Cu(111). However, the total impendence of Cu(100) was still larger than that of Cu(111) (6028 Ω·cm^2^ + 5417 Ω·cm^2^·s^−1/2^ > 3070 Ω·cm^2^ + 7071 Ω·cm^2^·s^−1/2^), which suggested a more protective ability of oxides on Cu(100).

After 10D corrosion, oxide with ~1 μm thickness was deposited on both SCs, but their properties varied widely. As Figure 10c,f showed, the interface between oxides and Cu(111) was defective, and electrolytes can penetrate the substrate through channels in the coarse oxides (bluish lines). Moreover, the interface between oxides and Cu(100) showed fewer defects and was more compact. The matched EC for Cu(100) was Figure 9c, where a compact oxides layer formed on Cu(100) SC, and the electrolyte hardly penetrated the substrate. All parameters were involved with the properties of the oxides layer, and the compact layer accounted for the largest capacitive arc in EIS. Figure 9d shows a typical two-time constant EEC for Cu(111) SC. The red lines showed the channel of the penetration, and R_ct_/C_dl_ symbolized the interface of S/M. It was noted that the C_dl_ of Cu(111) SC was one magnitude order larger than that of Cu(100) SC. Capacitance is a key parameter to evaluate the amount of water penetration. The amount of water penetration into the porous oxides affected the dielectric constant of the oxides. Since the dielectric constant of water is much higher than semiconductors. The dielectric constant of porous oxides increased, with electrolyte penetrating into oxides. Furthermore, the R_ct_ of Cu(100) exceeded R_ct_+R_o_ of Cu(111) at a larger margin (14,274 Ω·cm^2^ > 4117 Ω·cm^2^ + 5306 Ω·cm^2^). Thus, the compactness of the passive film on the (001) surface planes was greater than those on the (111) surface planes, which was consistent with the results of the microstructural analysis.

## 4. Conclusions

EIS with different prolonged durations was conducted, and various behaviors were detected in different corrosion stages. The following conclusions can be drawn:

In the initial stage, corrosion behavior was decided by air-formed oxide film instead of bare metal itself. The film on Cu(111) SC exhibited a higher possibility to dissolved due to the preferential adsorption of ${\mathrm{Cl}}^{-}$ on Cu(111) planes, which resulted in a smaller capacitive loop during the EIS test. After 24H, corrosion, dissolution and deposition occurred simultaneously. The dissolution process leads to the exposure of faceted (100) crystal planes macroscopically. The bottom of the indentation in Cu(100) SC was not flat, and it was shown as extensive rectangular steps microscopically. The inverted pyramids can be understood as the orthogonal side walls of one cubic corner on the Cu(111) SC surface. Both SCs showed straight lines in low frequency of Nyquist diagrams, which suggested an obvious mass diffusion process. After prolonged corrosion, compact oxides and less defective interfaces formed on Cu(100), which contributed to the good protection of oxides and a larger capacitive loop in Nyquist. Moreover, a defective interface with leaks and the uneven surface formed on Cu(111), and then the electrolyte penetrated the substrate through channels in the defective oxides.

## Figures and Tables

**Figure 1 materials-16-01740-f001:**
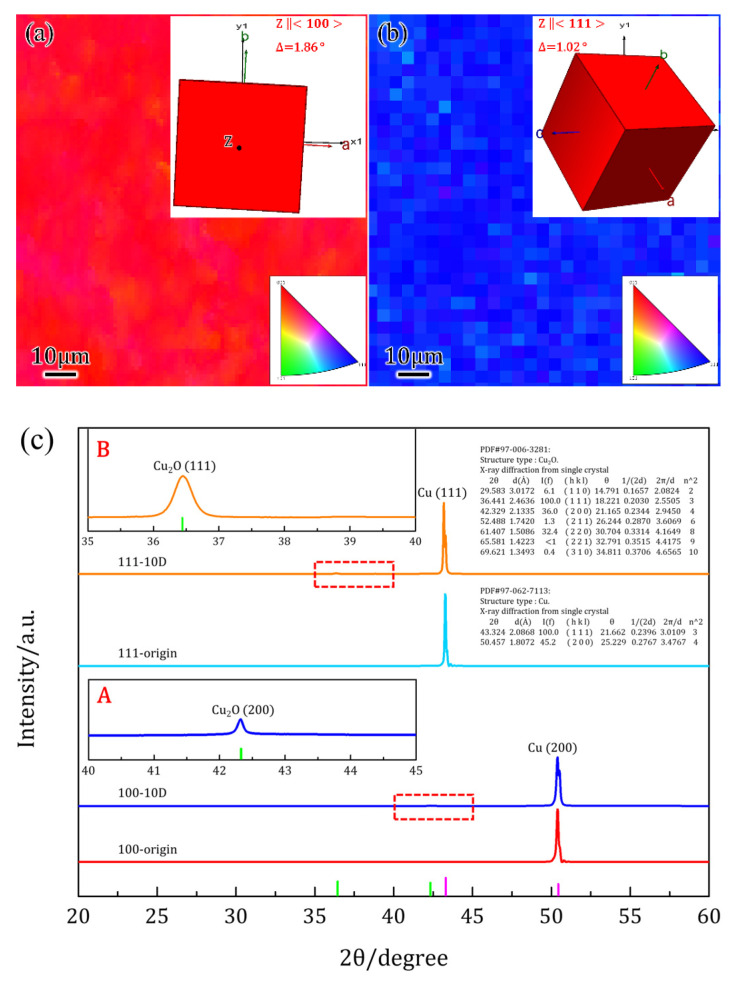
Orientation and constitution characterization of as-received and corroded Cu(100)/Cu(111) single crystals (SCs): (**a**,**b**) inverse pole pictures (IPFs) of as-received SCs, where the pure red and blue colors suggested the (100) and (111) planes, respectively, and the embedded unit cells symbolized the misorientation within 2°; (**c**) XRD of as-received and corroded SCs, where the emerged Cu_2_O(200) and Cu_2_O(111) peaks defined the constitution of corrosion products as Cu_2_O ((A) and (B) were enlarged views of Cu(100) -10D from 40° to 45° and Cu(111) -10D from 35° to 40° respectively).

**Figure 2 materials-16-01740-f002:**
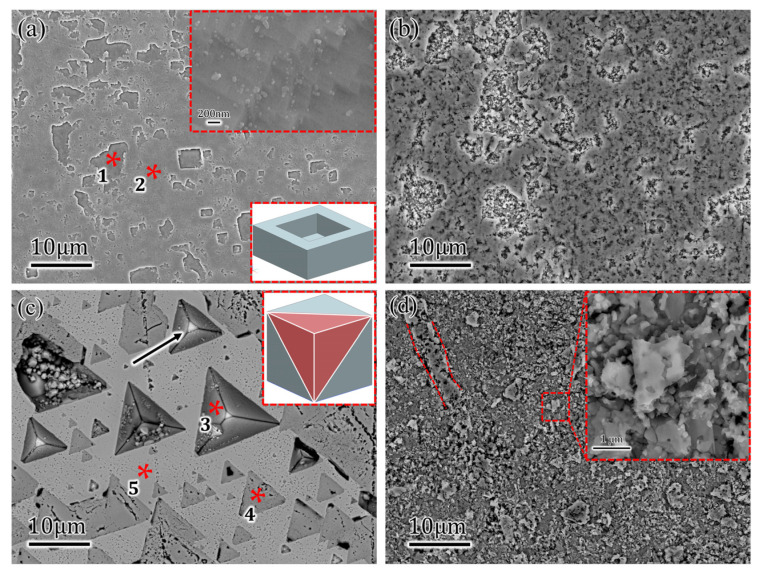
Top morphologies of Cu(100)/Cu(111) after 24 h (24H) and 10 days (10D) open circuit potential (OCP): (**a**) rectangle pits caused by dissolution on Cu(100) after 24H OCP (1* and 2* were EPMA characterization positions for Cu(100)); (**b**) rough flocculent-like and slightly flat oxides on Cu(100) after 10D OCP; (**c**) inverted pyramids and triangle oxides on Cu(111) after 24H OCP (3*, 4* and 5* were EPMA characterization positions for Cu(111)); (**d**) rugged oxides formed on Cu(111) after 10D OCP.

**Figure 3 materials-16-01740-f003:**
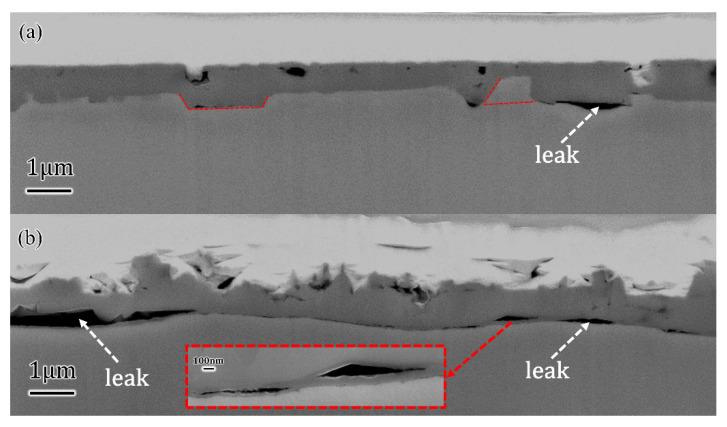
Cross-sectional morphologies of Cu(100)/Cu(111) after 10D OCP: (**a**) perfect interface and compact oxides on Cu(100) suggested well combination of oxides and Cu(100); (**b**) defective interface with long-rage leaks on Cu(111) suggested terrible combination of oxides and Cu(111).

**Figure 4 materials-16-01740-f004:**
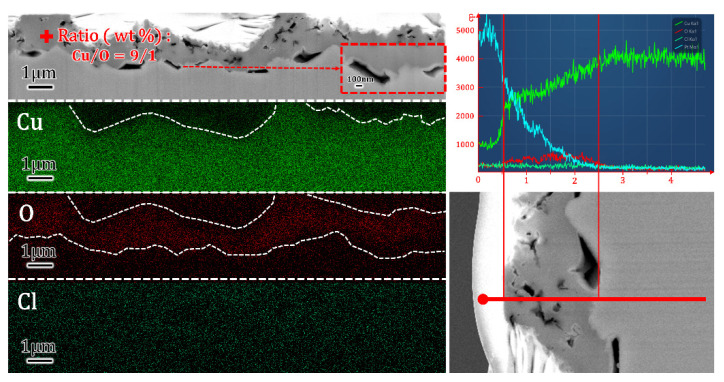
EDS of cross-sectional Cu(100) simple after 10D OCP. Point characterization showed the ratio of Cu/O was 9/1. Map of elements and line scan showed the oxide layer on substrate.

**Figure 5 materials-16-01740-f005:**
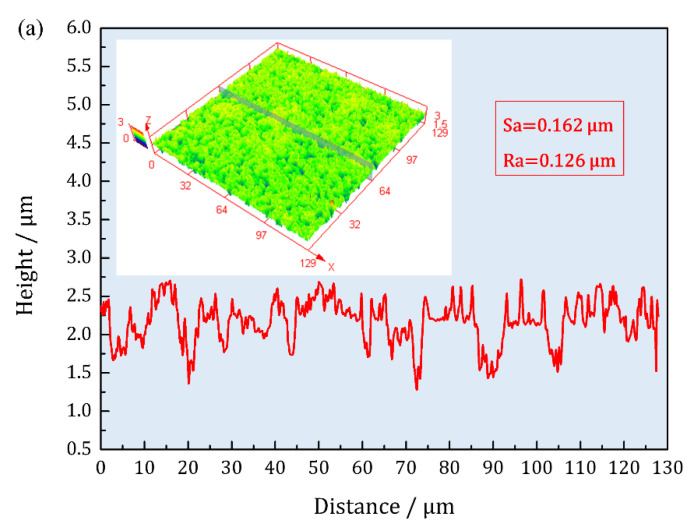
Laser-scanning confocal microscopy (LSCM) of corroded Cu(100)/Cu(111) after 10D OCP: (**a**) Sa = 0.162 μm and Ra = 0.126 μm for Cu(100) showed less rugged surface; (**b**) Sa = 0.189 μm and Ra = 0.154 μm for Cu(111) showed coarser surface (Sa stands for surface roughness, and Ra stands for linear rugosity).

**Figure 6 materials-16-01740-f006:**
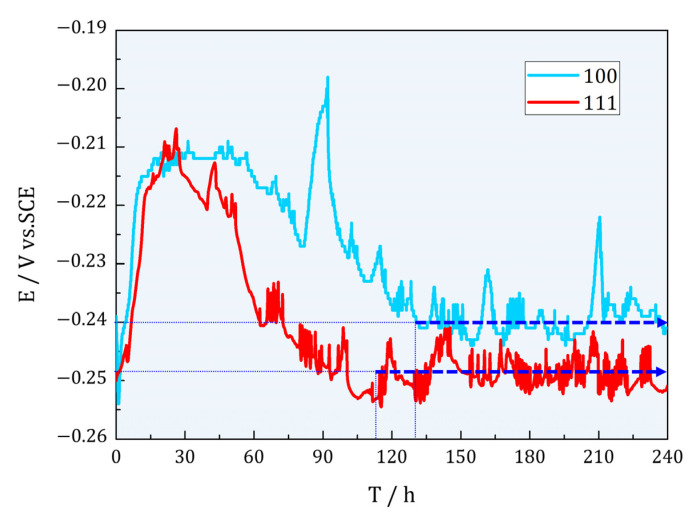
Ten days OCP of SC Cu(100)/Cu(111). Both curves increased abruptly to peaks and then decreased slowly to fluctuated values.

**Figure 7 materials-16-01740-f007:**
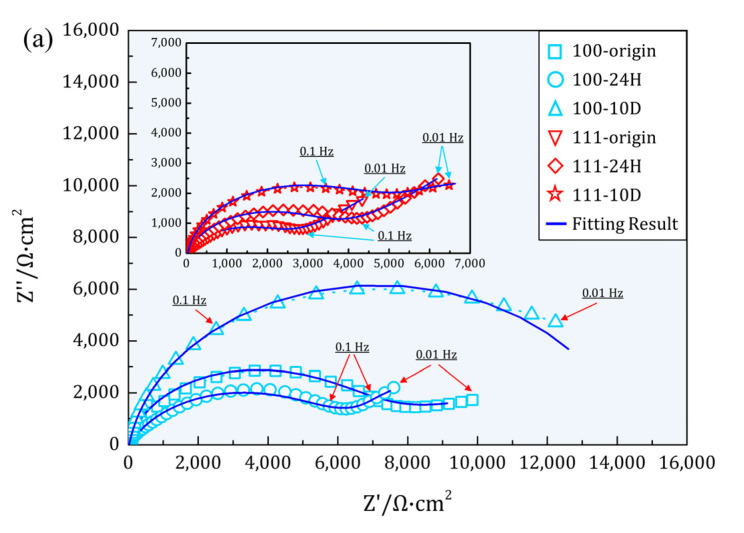
EIS of Cu(100)/Cu(111) in different corrosion stages: (**a**) Nyquist diagrams of Cu(100)/Cu(111); (**b**,**c**) f−|z| and f− phase angle, respectively (polygons represent the experimental results and blue lines represent the fitting results); (**d**) histogram of |Z| at f=0.01 Hz extracted from (**b**).

**Figure 8 materials-16-01740-f008:**
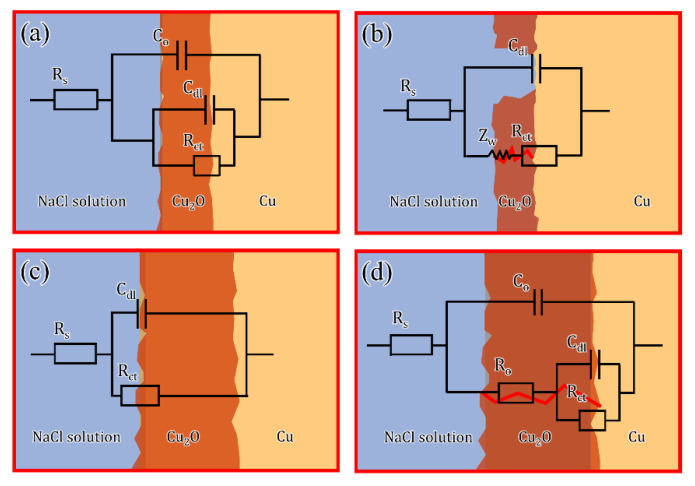
Equivalent electrical circuits (EECs) of Cu(100)/Cu(111) in different corrosion stages: (**a**) $R_{s}\left(C_{o}\left(C_{dl}R_{ct}\right)\right)$ for as-received Cu(100)/Cu(111); (**b**) $R_{s}\left(C_{dl}\left(Z_{w}R_{ct}\right)\right)$ for Cu(100)/Cu(111) after 24H OCP; (**c**) $R_{s}\left(C_{po}R_{po}\right)$ for Cu(100) after 10D OCP; (**d**) $R_{s}\left(C_{o}R_{o}\left(C_{dl}R_{ct}\right)\right)$ for Cu(111) after 10D OCP.

**Figure 9 materials-16-01740-f009:**
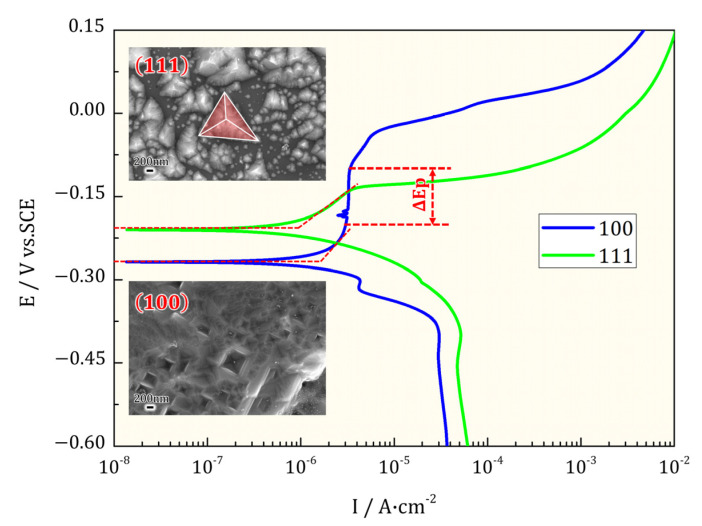
Potentiodynamic scan of Cu(100)/Cu(111): Cu(100) showed a narrow passivation rage; hierarchical corrosion steps and pyramids were found on Cu(100) and Cu(111), respectively.

**Figure 10 materials-16-01740-f010:**
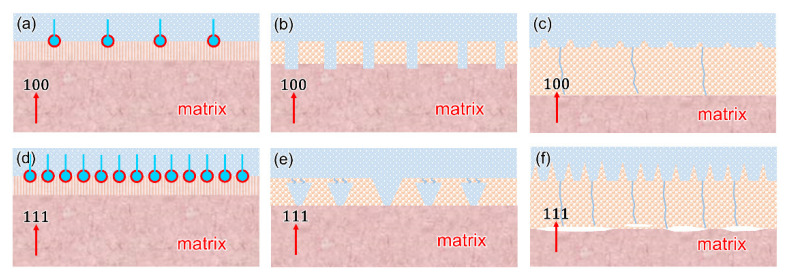
Corrosion mechanism of Cu(100)/Cu(111) in different corrosion stages: (**a**,**d**) more efficient adsorption of ${\mathrm{Cl}}^{-}$ on Cu(111) than Cu(100); (**b**,**e**) rectangle indentations on Cu(100) and inverted pyramids beneath oxides on Cu(111); (**c**,**f**) compact oxides and perfect interface on Cu(100) while defective interface with channels on Cu(111) after 10D OCP.

**Table 1 materials-16-01740-t001:** Atomic percentage at different positions on corroded surface after 24H and 10D OCP.

Sample		Cu	O	Cl
100	Position 1	95.9153	3.9486	0.1361
	Position 2	78.7958	20.9565	0.2477
111	Position 3	98.2529	1.6706	0.0765
	Position 4	69.6481	29.6041	0.7478
	Position 5	93.8035	6.1203	0.0763

**Table 2 materials-16-01740-t002:** Parameters extracted from EECs in different corrosion stages.

Parameters	100			111		
	0H	24H	10D	0H	24H	10D
R_S_ (Ω·cm^2^)	18.15	15.96	21.91	15.74	15.87	22.31
C_dl_ (μF·cm^−2^)	5.89 × 10^−5^	2.04 × 10^−5^	2.67 × 10^−4^	6.09 × 10^−5^	3.55 × 10^−5^	3.2 × 10^−3^
n_dl_	0.62	0.75	0.91	0.71	0.79	0.89
R_ct_ (Ω·cm^2^)	4548	6028	14,274	2832	3070	4117
C_o_ (μF·cm^−2^)	1.26 × 10^−5^			1.83 × 10^−5^		1.75 × 10^−4^
n_o_	0.9			0.65		0.89
R_o_ (Ω·cm^2^)						5306
Z_w_ (Ω·cm^2^·s^−1/2^)		5417			7071	
Chi-Square	0.0074	0.0387	0.0090	0.0188	0.0203	0.0104

**Table 3 materials-16-01740-t003:** Abbreviations of EECs of Cu(100)/Cu(111) after different prolonged OCP.

SC	0H	24H	10D
100	$R_{s}\left(C_{o}\left(C_{dl}R_{ct}\right)\right)$	$R_{s}\left(C_{dl}\left(Z_{w}R_{ct}\right)\right)$	$R_{s}\left(C_{po}R_{po}\right)$
111	$R_{s}\left(C_{o}\left(C_{dl}R_{ct}\right)\right)$	$R_{s}\left(C_{dl}\left(Z_{w}R_{ct}\right)\right)$	$R_{s}\left(C_{o}R_{o}\left(C_{dl}R_{ct}\right)\right)$

## Data Availability

Not applicable.

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
