# Peer review of "Electrochemical Impedance Spectroscopy (EIS) Explanation of Single Crystal Cu(100)/Cu(111) in Different Corrosion Stages"

_materials, 2023, doi:10.3390/ma16041740_

Round 1
Reviewer 1 Report
Electrochemical anaysis was examined to Cu plates for analyzing the corrosion properties. Some revision is required to publish in this journal.
1. In exprimental method, the author was used the SC Cu(100)/Cu(111) plate, the specific chemical composition is required.
2. English correction is required to publish by expert. Some words are difficult for readers to understand.
3. The author was used the 3.5% NaCl solution. 3.5 vol.% or 3.5 wt.% ?
4. What is the mean of "ultrapure water" in the Materials and experimental?
5. What is the reference voltage of the SCE electrode?
6. In Fig. 4, the cross-sectional mapping images are difficult to distinguish the specific elements.
Author Response
Dear reviewer:
I am grateful for your rigorous and professional supervisions. Revision was conducted by author and colleagues. Some explanations were summarized as follows according to your review:
- Chemical composition of SC Cu(100)/Cu(111) plate was confirmed by supplier. The purity of Cu was more than 99.99% (mass%). This information was also added to the description of materials.
- Some typos and grammatical mistakes in script were revised carefully and manuscript was also checked by a native English-speaking colleague.
- “3.5 wt.% NaCl solution” should be used consistent with other researches for comparison purpose.
- “Ultrapure water” was replaced by “deionized (DI) water” with resistance > 18.2 MΩcm according to reference [35].
- The reference voltage of KCl-saturated calomel electrode (SCE) electrode was 0.2438 V (vs SHE, Standard Hydrogen Electrode).
- It was a pity not to supply a mapping image with enough resolution in Fig. 4. Based on this condition, the outline of element enriched area was highlighted to distinguish the segregation of elements.

Reviewer 2 Report
The article titled “In-suit Electrochemical Impedance Spectroscopy (EIS) Explanation of Single Crystal Cu(???)/Cu(???) in Different Corrosion Stages” by Lin et al. is an interesting work that describes how crystal orientation plays a vital role in improving the corrosion resistance. There are still some questions that need clarification and suggestions to strengthen the manuscript further:
1. The abstract needs to be improved to include a little more background and motivation for the work. The authors state that “Combined with equivalent circuit (EC) diagrams, corrosion mechanism of Cu(100)/Cu(111) in different stages was proposed”. However, it is suggested that the authors include what this proposed mechanism is in a compact version within the abstract. Comparison between the capacitive loops between Cu(100) and Cu(111) is also a little vague and it is suggested to also add the values of impedance as measured by EIS to provide evidence to improvement in corrosion resistance. Please also proofread the abstract for grammar and typos (In-situ, not In-suit for example).
2. The manuscript doesn’t have line numbers which makes it difficult to reference a particular line. Citations should be arranged in the proper order to avoid jumping from one reference to the other.
3. Significant revisions are needed for the Introduction section where grammar and typos need to be addressed. Also, the way it is written is somewhat jumping from one idea to another where clearly the flow of manuscript is missing. There is a disconnect between the background, corrosion and EIS. A suggestion is that the authors should start by introducing the role of copper in different applications, followed by how corrosion prevents such applications from fully utilizing copper as a material and then introduce the way such corrosion can be quantified through EIS with a stark comparison between the different crystal orientation. All these ideas should be fully supported by literature (which I believe is already there in the form of citations in the manuscript) with references in the proper ascending order.
4. It is claimed that in Fig 2, (a), (b) and (c), (d) shows the fairly pure and oxidized forms of Cu(100) and Cu(111) respectively. However, the two figures at 24H OCP and 10D OCP are much different for both the species. It is understood that given the (100) orientation, it is fairly easy for the oxidized forms of Cu particles to hide the morphology. However, it is interesting to see that the pyramidal structures of Cu(111) are totally lost in figure 2(d). It is suggested that this explanation be added to the Results section along with the observations from figure 2.
5. Page 7, paragraph 2 seems like should be in the Introduction section. The authors can use the Yang and X.Y. references to justify their results but it doesn’t need all the background all over again.
6. Table 1 can be moved up to page 8 from page 10 along with its first mention on page 8.
7. All figures should be consistent in terms of their design. Figures 3 and 4 have red dotted border whereas other figures have solid borders. Also, the plots in figures 4 and 5 should have a better resolution in order to see the axis labels and values properly.
8. It is suggested that the authors also add the morphological and EDS data for the 24H OCP to show the change in corrosion over a period of time as quantified.
9. For figure 5, it is understood that Sa is surface roughness but the manuscript doesn’t state what Ra stands for. It is suggested that this information be also added within the captions of the figure.
1. For figure 7, it is suggested that the number of symbols be reduced especially for the lower plots to have a clear observation. For 7A, Cu(111), it is observed that the imaginary impedance is lowest for the original sample, a little higher for the 24H and highest for 10D sample, which looks good. However, for the Cu(100), why does the original sample have a higher impedance than the 24H seeing that over the period of 24H, the original should have improved its impedance?
1. It is suggested that Figure 7B should have some sort of offset to isolate the two copper species to read the observations correctly.
1. It is seen that there are consistent typos in the entire manuscript for words such as “In-situ”, “range” and “samples”. It is suggested to thoroughly check the manuscript for grammatical and spelling errors.
1. The conclusion can be in a single paragraph rather than three individual points.
Author Response
Dear reviewer:
I am grateful for your rigorous and professional supervisions. Revision was conducted by author and colleagues. Some explanations were summarized as follows according to your review:
- Some backgrounds and one reasonable motivation for this work were provided in the abstract. The corrosion mechanism of Cu(100)/Cu(111) in different stages was complemented and the values of impedance as measured by EIS were also supplied to illustrate the higher capacitive loops of Cu(100) after 10D OCP.
- Line numbers has been added to manuscript and citations were also rearranged in ascending order.
- Grammatical errors and typos in the Introduction section were revised carefully and completely. As reviewer said, it is somewhat jumping from one idea to another. Combining the flow provided by reviewer, Introduction was revised significantly and the following thread of thought may be also reasonable:
wide usage of copper in marine environments due to excellent corrosion resistance → disintegrating the corrosion process into four parts → anisotropy exist widely in dissolution and deposition stage (influencing the corrosion kinetics) → EIS is a powerful tool in the investigation of the corrosion behavior → in-situ EIS in this paper can remain the corrosion frontier → single crystals were adopted to exclude the effects of microstructures.
- To explain the significant variation, Fig 2 (d) was replaced by another similar picture and an enlarged picture was also displayed. Form the vague tripe (surrounded by red dotted lines) and enlarged morphology in Fig 2(d), the flakes on Cu(111) might be the remained surface as the flat areas in Fig 2(b). It was deduced that more extensive dissolution occurred on Cu(111) and the pyramidal structures were covered totally by corrosion products.
- The Yang and X.Y. references in page 7 was abbreviated to one sentence, which provided necessary information to certify the reliability of experimental results.
- Table 1 has been moved to page 8 after the paragraph where it was first mentioned.
- Figures 3 and 4 have been designed consistent with other figures, where red dotted border was removed and the axis labels and values were displayed clearly.
- The morphology of 24H OCP has been showed in Fig. 2(a) and (c), where rectangular indentations and inverted pyramids for Cu(100) and Cu(111) respectively. Due to the slight oxidation for 24H OCP, electron probe microanalysis (EPMA) was adopted to detect the component at different positions, which can supply more precise data than EDS in quantity. The EPMA results were showed by table 1 and corresponding datum were analyzed in manuscript.
- Ra stands for linear rugosity. It was also added to the results and the captions of Fig 5.
- Size of symbols and width of lines in Fig. 7 were adjusted to make it easier to distinguish two copper species correctly. Furthermore, the phenomenon that the original sample have a higher impedance than the 24H for Cu(100) can be explained in this way: dissolution process led to plenty of pits on Cu and the side wall and bottom of pits were almost bare substrate according EPMA results at position 1. That is to say, air-formed film was destroyed and more bare copper was exposed in electrolyte, which controbuted to the decrease of imaginary impedance after 24H OCP. While for Cu(111), corrosion occurred in another way that only partial pits were exposed directly and most dissolution occurred through leaks as Fig. 2(c) showed. This may compensate for the decrease of impendence caused by dissolution.
- Size of symbols and width of lines in Fig. 7 were adjusted to make it easier to distinguish two copper species correctly.
- “In-situ”, “range” and “samples” have replaced the original typos and similar grammatical and spelling errors were also revised carefully.
13. Conclusion has been converted into a single paragraph.

Reviewer 3 Report
This work investigated the corrosion behavior of single crystal Cu(???)/Cu(???) in different corrosion stages by in-situ EIS. Some issues listed in the following points should be clarified and supplemented before publication:
1. The peaks in XRD indicates crystal planes, not the directions. The author mentioned the SC Cu(100)/Cu(111) plates supplied by Corporation, but Cu(200) and Cu(111) peaks were observed in Fig. 1. Please explain this result. Additionally, where is Fig. 1(c)? Standard PDF card along with index is necessary.
2. The word “simple” or “Sample”?
3. It is suggested to add the microstructure of as-received Cu(100)/Cu(111) single crystals.
4. In Fig. 4, the oxide is defined to Cu2O by EDS. It’s not proper because EDS is only capable of qualitative analysis. The oxide should be identified by XPS.
Author Response
Dear reviewer:
I am grateful for your rigorous and professional supervisions. Revision was conducted by author and colleagues. Some explanations were summarized as follows according to your review:
- To certify the orientation of single crystals, electron backscattered diffraction (EBSD) of as-received plates were added. As the inverse pole figure (IPF), unit cells and XRD results in Fig. 1 showed, the preferential orientation of SCs were that normal directions were parallel to 〈100〉 and 〈111〉 direction respectively. The index Cu(200) was defined according to PDF cards (also showed in Fig. 1), which is the result of extinction law of X-ray diffraction. Fig. 1(c) was replaced by the XRD of as-received and corroded samples. Standard PDF cards of Cu and Cu2O were added to Fig. 1 to define the component.
- “Samples” should be used and similar typos were also revised.
- The as-received single crystals were polished to mirror-like surface and there was not valuable information from the microstructure.
- It was commendable to point out the imprecision of EDS in qualitative analysis and XRD of corroded samples were supplied to explain the composition of corrosion products due to the enough thickness of oxides.

Round 2
Reviewer 2 Report
The authors have made major modifications to the manuscript as per the suggestions. I do not have any further comments.
Author Response
Dear reviewer:
Thank you very much for your positive assessment.
Best wishes!
Reviewer 3 Report
The revised manuscript can be accepted.
Author Response

(The authors gave the same response as above.)
